# Snow Avalanche Impact Measurements at the Seehore Test Site in Aosta Valley (NW Italian Alps)

**Margherita Maggioni** [1,2,*], **Monica Barbero** [1], **Fabrizio Barpi** [1], **Mauro Borri-Brunetto** [1], **Valerio De Biagi** [1], **Michele Freppaz** [2], **Barbara Frigo** [1], **Oronzo Pallara** [1] **and Bernardino Chiaia** [1]

1   Department of Structural, Geotechnical and Building Engineering, Politecnico di Torino; Corso Duca degli Abruzzi 24, 10129 Torino, Italy; monica.barbero@polito.it (M.B.); fabrizio.barpi@polito.it (F.B.); mauro.borri@polito.it (M.B.-B.); valerio.debiagi@polito.it (V.D.B.); barbara.frigo@polito.it (B.F.); oronzo.pallara@polito.it (O.P.); bernardino.chiaia@polito.it (B.C.)

2   Department of Agriculture, Forest and Food Sciences and NatRisk, Università di Torino; L.go Braccini 2, 10095 Grugliasco (TO), Italy; michele.freppaz@unito.it

\*   Correspondence: margherita.maggioni@unito.it

**Abstract:** In full-scale snow avalanche test sites, structures such as pylons, plates, or dams have been used to measure impact forces and pressures from avalanches. Impact pressures are of extreme importance when dealing with issues such as hazard mapping and the design of buildings exposed to avalanches. In this paper, we present the force measurements recorded for five selected avalanches that occurred at the Seehore test site in Aosta Valley (NW Italian Alps). The five avalanches were small to medium-sized and cover a wide range in terms of snow characteristics and flow dynamics. Our aim was to analyze the force and pressure measurements with respect to the avalanche characteristics. We measured pressures in the range of 2 to 30 kPa. Though without exhaustive measurements of the avalanche flows, we found indications of different flow regimes. For example, we could appreciate some differences in the vertical profile of the pressures recorded for wet dense avalanches and powder ones. Being aware of the fact that more complete measurements are necessary to fully describe the avalanche flows, we think that the data of the five avalanches triggered at the Seehore test site might add some useful information to the ongoing scientific discussion on avalanche flow regimes and impact pressure.

**Keywords:** snow avalanches; impact forces; experimental data; full-scale test site

---

## 1. Introduction

Snow avalanche experimental test sites have existed for a long time in Europe and in other countries, both at laboratory and at full scale. An overview of the European avalanche test sites can be found in [1,2], but additional literature exists regarding each specific test site (e.g., [3–5]). The main goal of a full-scale test site is to measure dynamic variables within an avalanche flow from release to runout, in order to characterize the different avalanches and investigate their dynamics. Rich and consistent databases of real avalanches allow one to test dynamical models to simulate the phenomena or at least to find empirical rules to describe them and their effects (e.g., [6]).

In particular, avalanche impact pressure measurements are of extreme importance in issues such as hazard mapping and the design of buildings or infrastructures exposed to avalanches. In the European test sites, structures such as pylons, plates, or dams have been used to measure impact forces and pressures induced by avalanches.

Impact pressure measurements, combined with other data when possible (velocity, density, temperature, flow depth), give information on the avalanche flow regime. Recently, the development of more sophisticated instruments makes it possible to look inside the avalanche flow and analyze the

different flow regimes [7]. However, even from the simple analyses of avalanche deposits, it emerges that avalanches are characterized by different flow regimes [8].

More difficult is determining the effects of the impact of avalanches with different flow regimes on structures. Johannesson et al. [9] summarized the most recent findings regarding the impact of avalanche flows of different types against dams. At the two extremes, consider the effect of powder and wet dense snow avalanches: Their impact on buildings clearly produces completely different damages. Apropos, in terms of engineering, as an example in Switzerland (and also in other European countries), the structural design of a building, which can be impacted by a powder snow avalanche, is made following the guidelines related to the wind action [10], while for a dense flowing avalanche, the reference is different [11], emphasizing two completely different approaches. However, a dense flowing avalanche itself can be of different types: Simply wet or dry as in [12] or showing different flow regimes.

The importance of considering different flow regimes in avalanche hazard mapping has been analyzed by [13] and some recent works are going into the direction of creating different hazard scenarios with respect to the snow conditions [14].

In this paper, we present different measurements made at the Seehore test site in Northwestern Alps in Italy [15,16] with particular focus on impact forces recorded on an instrumented obstacle placed along the avalanche path. The aim of the paper is to present and discuss the force measurements in combination with other characteristics of the five selected avalanches, in order to contribute to the ongoing scientific discussion regarding impact pressures and flow regimes.

## 2. Materials and Methods

### 2.1. Seehore Test Site

The test site, called Seehore, is located in Aosta Valley in the Northwestern Italian Alps (45°51′10″ N; 7°50′30″ E) and was operative in the period 2009–2016.

The slope, with an elevation difference of about 300 m (from 2300 to 2570 m asl), has a mean slope angle of about 28° and an NNW aspect. Spontaneous and artificial avalanches were observed, the latter being released on a routine basis to secure the ski runs, as the site is located within a ski resort (Monterosa Ski): The threshold for attempting an artificial release was around 30 cm of new snow. Hence, the artificially released avalanches were generally small to medium-sized avalanches [17]: The release volume was typically about 200–400 m$^3$, but could reach about 2000 m$^3$ for thick slab release. Even spontaneous releases, generally out of the opening season of the ski-resort, occurred and were mainly wet-snow avalanches. In the vicinity of the test site, the automatic weather station (AWS) of Gressoney-L.T.–Gabiet (2379 m asl) is located, which provides data on air temperature and snow depth.

For a complete description of the test site, the field and remote sensing measurements and the instrumented obstacle placed along the avalanche path refer to [15,16], while in the next section, we briefly report some details about the instrumented obstacle, which are useful for the current paper.

### 2.2. Instrumented Obstacle

The avalanche test site was equipped with an instrumented obstacle, which measured the effects of avalanches impacting on it. The structure of the obstacle, located at 2420 m asl on a slope with inclination of about 35°, was made of galvanized steel profiles (4.0 m high, protruding 2.7 m from the natural slope profile). It consisted of an upper part carrying the sensors and directly exposed to the avalanche impact, which was connected with a bolted connection with a predefined strength to a lower section bolted to the concrete foundation and serving as a support.

The impact surface was made of an array of 5 grooved aluminum vertical plates (1110 mm wide, 180 mm high) placed at different heights. Their position could be changed since each plate was supported by two load transducers, mounted on slides that could be easily moved along vertical guides. When the plates were mounted adjacently, the total impact area was 1.133 m$^2$. During the first season (2010–2011), one plate was mounted isolate closer to the ground (at 70 cm) and the other

four were mounted adjacently at a higher level from the ground (lowest plate at 120 cm). After the experience of the first season, all five plates were mounted adjacently at a distance from the ground (for the lowest plate) of 50 cm. The configurations of the plates for the 5 selected avalanches will be shown and discussed hereafter (Section 2.3).

The obstacle was equipped with several sensors that measured different parameters: The impact force, the acceleration of the structure itself, the air temperature, and the atmospheric pressure. In particular, the impact force was measured at a sample rate of 2000 Hz by 10 transducers HBM U10M with nominal maximum load of ±5, ±12.5, and ±25 kN, and accuracy of 0.2%. At the beginning (winter season 2010–2011), the recording of the measurements at the obstacle was manually triggered from a computer placed in a room located 600 m further from the test site, connected to the obstacle by means of an optical-fiber line. Starting from November 2011, the recording of an event data was automatically triggered from the accelerometers placed on the structure itself when a certain threshold level was attained. Therefore, from the winter season 2011–2012, the impacts of spontaneous avalanches were also recorded and analyzed.

As described in the following paragraph, in some cases, some pressure sensors were not operative. This condition could be easily detected through the analyses of the recorded signals, which showed a typical wave form of an open electrical circuit.

### 2.3. Selected Avalanches

Among all the avalanches that occurred at Seehore, the five avalanches considered in this work were artificially triggered or spontaneously released on 7 December 2010, 5 March 2011, 17 April 2013, 20 January 2014, and 30 April 2014. The selection of these 5 events was made according to the available measurements and in order to cover different ranges of force measurements and avalanche types. In particular, a powder cloud developed during the events of 7 December 2010 and 5 March 2011, while the event of 20 January 2014 was a small dry dense avalanche; the events of 17 April 2013 and 30 April 2014 were spontaneous wet snow avalanches. After each event, a field campaign was made to get information on the avalanche extension and characteristics; for some events, laser scanning and photogrammetry were also made in order to obtain information about the mass balance and the front velocity of the avalanches. The five selected avalanches result to be the most well documented in the database.

For the five selected avalanches, the load transducers on the obstacle were mounted with the different configurations shown in Figure 1, which also highlights the inoperative load transducers.

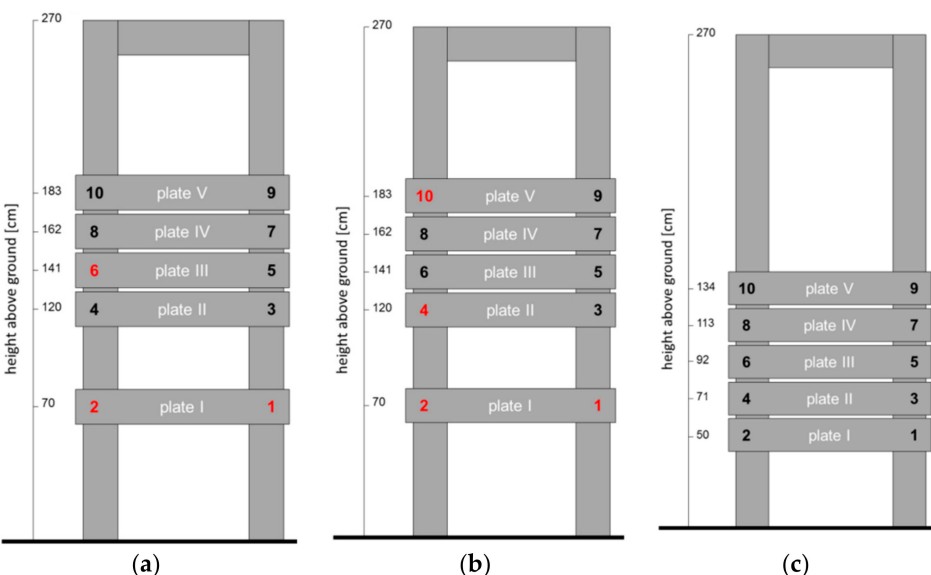

**Figure 1.** Configuration of the obstacle for the five selected avalanches: (**a**) 10 December 2010, (**b**) 5 March 2011, (**c**) 17 April 2013, 20 January 2014, and 30 April 2014. The red numbers indicate not-working load transducers.

*2.4. Data Analyses*

For the 5 selected events, we computed the total force time-history on the obstacle simply by summing the measured forces recorded by each load transducer. As soon as some transducers were sometimes not operative, a simple extrapolation scheme was adopted, assuming for the malfunctioning instrument a force value corresponding to the average value of the measurements of the others. In particular, for the events on 10 December 2010 and 5 March 2011 (when the defective transducers were those depicted in Figure 1) the total force was calculated, on the plates II-V that were impacted by the avalanches, as the sum of the forces measured by the operative transducers (N = 7 for 2010 and N = 6 for 2011) divided by N and multiplied by 8 (transducers mounted on the four considered plates).

As in the literature it is more common to discuss impact pressure, which is the dynamical variable used to analyze the effects of an avalanche, we calculated the total impact pressure, simply dividing the total force by the impacted area. When all load transducers were operative, the total impact area was 1.133 m$^2$ (Figure 1c); in the cases of 7 December 2010 and 5 March 2011, the impacted area was 0.9 m$^2$ (Figure 1a,b).

We also analyzed the vertical distribution of the pressures on various plates (impact area of 0.1998 m$^2$, each) and evaluated (from the force measurements averaged with a moving mean over 0.2 s) the vertical profile of the impact pressures for the duration of the impacts. We chose to make the moving average over 0.2 s to reproduce the mean evolution of the pressure. In order to calculate the impact pressure on a plate where a transducer was not operative, we doubled the value of the operative transducer. For some elaborations, a low-pass filter set at 45 Hz was applied to the data in order to remove power supply interferences.

Moreover, for the events recorded on 17 April 2013 and 20 January 2014, when all the transducers were functioning and impacted by the flows, we also analyzed the time evolution of the position of the centroid of forces on the impact area, i.e., the position of the point of application of the resultant force on the area defined by the impact plates.

## 3. Results

*3.1. Avalanches Description*

Figure 2 shows the outlines of the five selected avalanches, while Table 1 reports their main characteristics. In the following paragraphs, we briefly describe each of them, in order to provide to the reader with a useful background to understand the paragraph dedicated to the measurements made at the obstacle (Section 3.2). In the following, details about the total forces are provided. These refer to the raw measurements, i.e., the data recorded by the datalogger, preliminary to any smoothing or data manipulation operations.

**Table 1.** Characteristics of the five selected avalanche events. Nomenclature as used in [8].

| | 7 December 2010 | 5 March 2011 | 17 April 2013 | 20 January 2014 | 30 April 2014 |
|---|---|---|---|---|---|
| triggering mechanism | artificial | artificial | spontaneous | artificial | spontaneous |
| manner of starting | slab | slab | slab | slab | loose-snow |
| number of triggerings | 1 | 2 | 0 | 2 | 0 |
| form of movement | mixed | mixed | dense | dense | dense |
| fracture depth (m) | 0.5 | 0.5 | 1 | 0.2 | // |
| secondary releases | Yes | yes | // | no | // |
| liquid water in snow | dry | dry | wet | dry | wet |
| surface layer snow density (kg/m$^3$) | 130 | 270 | 390 | 60 | // |
| deposition snow density (kg/m$^3$) | 300 | // | 600 | 180 | 300 |
| snow depth (m) * | 1.13 | 1.29 | 1.05 | 1.29 | 1.69 |
| daily air temperature (°C) * | 0.6 | −4.0 | 7.0 | −3.8 | 0.4 |
| max impact force – raw data (N) ** | 11,512 | 27,674 | 18,734 | 2158 | 7051 |
| max pressure – raw data (kPa) ** | 12.8 | 30.7 | 16.5 | 1.9 | 6.2 |
| number of operative load cells | 7/10 | 6/10 | 10/10 | 10/10 | 10/10 |
| front velocity at the obstacle (m/s) | 18–24 | 18 | // | 6–8 | // |
| snow wedge upwards the obstacle | no | yes | yes | yes | no (blocks) |
| snow wedge density (kg/m$^3$) | // | 320–360 | 420 | 220 | (400) *** |

* Parameters from AWS Gressoney-L.T.–Gabiet (2379 m asl). ** Positive values represent compressive forces.
*** Density of the blocks.

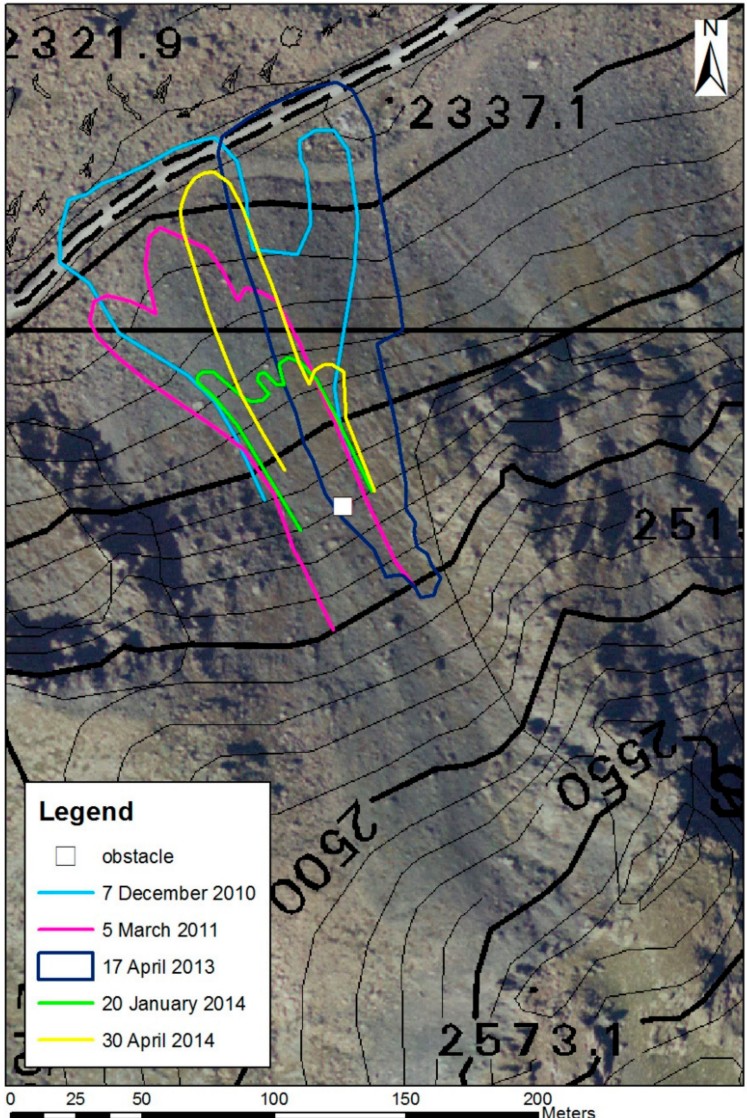

**Figure 2.** Outlines of the five selected avalanches in the track and deposition zones. Only for the 17 April 2013 event could we also determine the release zone well, and therefore the blue polygon shows the overall avalanche outline.

3.1.1. The Avalanche on 7 December 2010

On 7 December 2010, a dense dry snow slab avalanche was artificially released at about 2560 m asl; during the flow, a substantial powder component developed. Secondary releases, at about 2450 m asl, occurred while the avalanche was flowing (Figure 3). This event was one of the largest recorded at the test site in terms of release volume and runout distance and covered the ski run at the toe of the slope for a length of about 80 m, with a maximum deposition depth of about 2 m (Figure 2). The obstacle recorded two distinct impacts each about 1 s long and 2 s apart. The corresponding maximum total forces (calculated considering all the sensors placed on the obstacle as in Figures 1 and 4) were about 11,500 N and 8150 N, respectively (raw data).

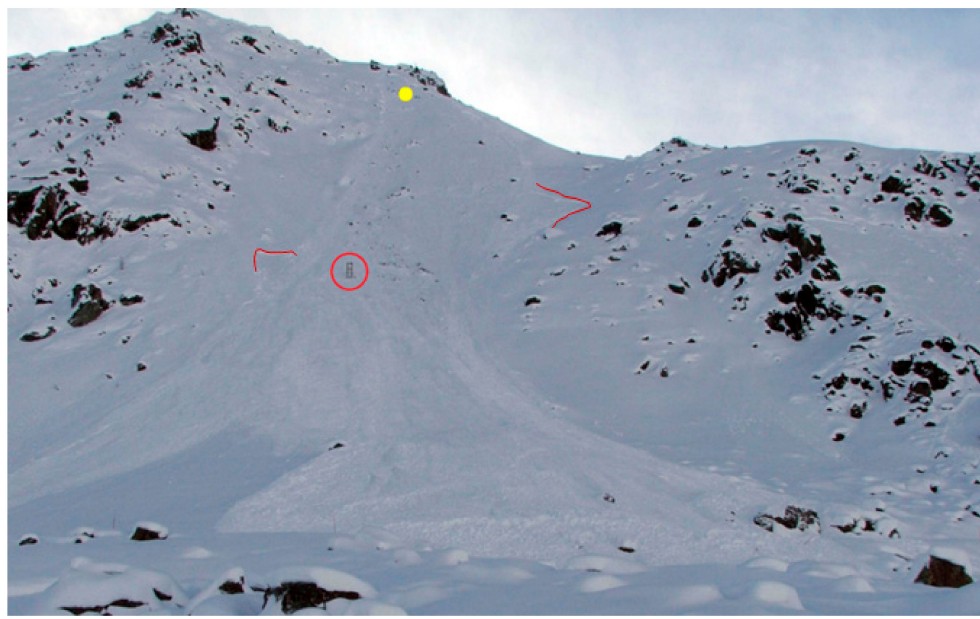

**Figure 3.** Overview of the 7 December 2010 avalanche. The yellow dot indicates the triggering point and the red circle the obstacle; the red lines highlight the secondary releases. Photo L. Pitet.

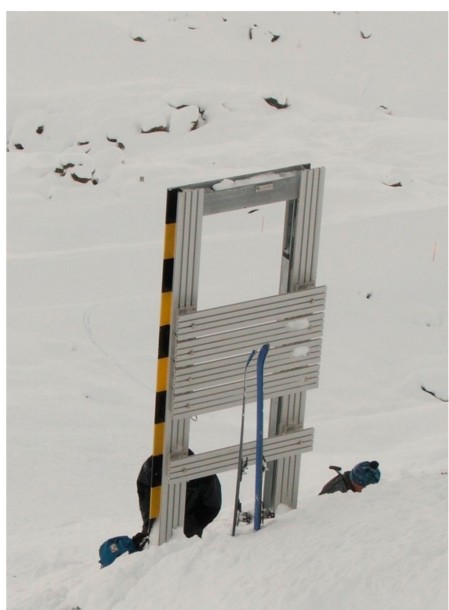

**Figure 4.** The obstacle after the 7 December 2010 avalanche (photo L. Pitet).

### 3.1.2. The Avalanche on 5 March 2011

On 5 March 2011, a first small dry snow slab avalanche was artificially released on the right side of the slope (Figure 5). During the motion, a powder component also developed, which flowed farther than the dense part that stopped in the deposition zone at about 2350 m asl. A small secondary release was triggered from the first flow at around 2460 m asl on the left side from the main avalanche flow, without reaching the obstacle. After those two flows had stopped, a second slab avalanche was artificially released from the top, at about 2570 m asl, which flowed straight down over the deposit of the previous avalanches impacting the obstacle, but the measuring system was not activated again and did not record impact forces. The final deposition (the three avalanches together) had a tri-lobe shape with an overall maximum width of about 55 m (Figure 2). The force measurements strongly fluctuated, in particular during the first 2 s of the impact; the maximum value of the total force (calculated

considering all the sensors placed on the obstacle as in Figure 1) was 27,650 N (raw data). Upstream the obstacle, a snow wedge originated, with a density ranging between 320 and 360 kg/m$^3$ (Figure 6).

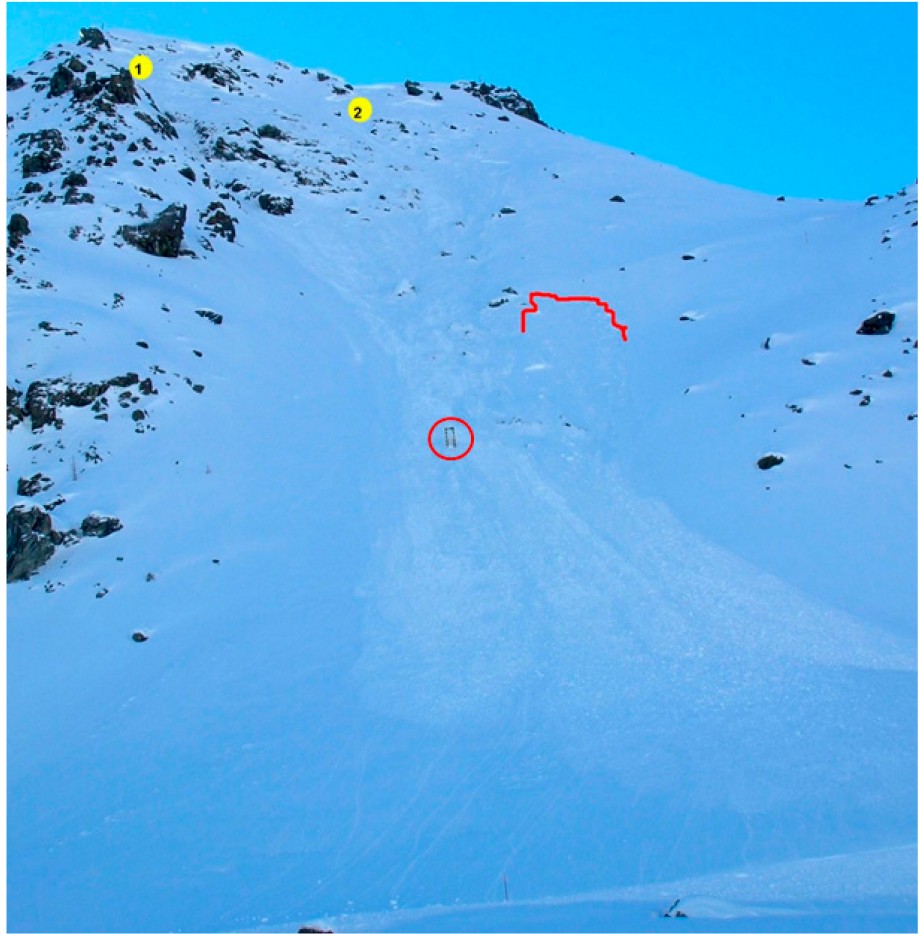

**Figure 5.** Overview of the 5 March 2011 avalanche. The yellow dot indicates the triggering points and the red circle the obstacle; the red line highlights the secondary release induced by the first flow. Photo M. Freppaz.

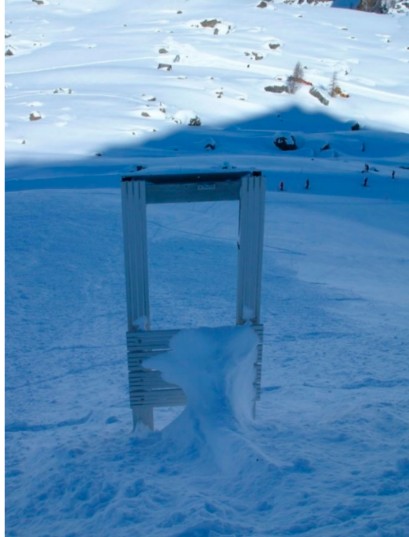

**Figure 6.** Obstacle after the events on 5 March 2011. Photo M. Freppaz.

### 3.1.3. The Avalanche on 17 April 2013

In the afternoon of 17 April 2013, a spontaneous avalanche occurred at the test site (Figure 2). It was observed by the ski resort personnel; a field campaign was made in the following days (Figure 7).

The weather conditions were typically spring-like, with air temperatures above zero also overnight from 15 April 2013 and a 15 cm snow cover melting in three days. On 17 April, the 0 °C isotherm was at 3800 m asl and the maximum air temperature was 14 °C (AWS Gressoney-L.T.–Gabiet). On 19 April, a survey was made only in the avalanche runout zone, for safety reasons. The slab avalanche released from just above the obstacle, where a fracture line was well visible (Figures 7 and 8), and overflew the ski-run. The deposit was made of wet agglomerates with an average density of 600 kg/m$^3$. The obstacle was automatically triggered at 17:30:07 UTC and recorded the impact, with forces (calculated considering all the sensors placed on the obstacle as in Figure 1) up to 18,500 N (raw data). The specific measurements around the obstacle were made on 24 April (Figure 9). The average density of the snow wedge formed by the avalanche against the obstacle was 420 kg/m$^3$. Traces of another small loose-snow avalanche were observed on the slope; it released from the ridge but remained on the left of the obstacle, without colliding.

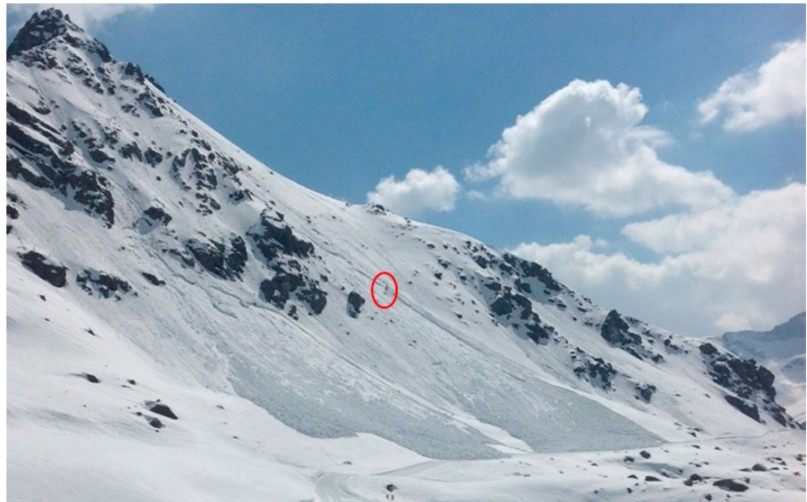

**Figure 7.** Overview of the 17 April 2013 avalanches. The red circle highlights the obstacle. Photo A. Welf.

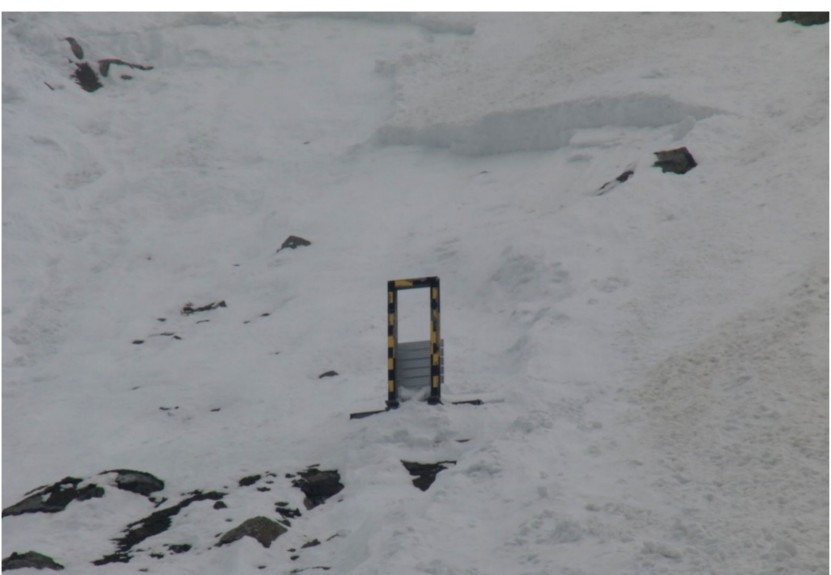

**Figure 8.** View of the obstacle from below after the event on 17 April 2013. The fracture line of the released slab is clearly visible. Photo L. Pitet on April 19.

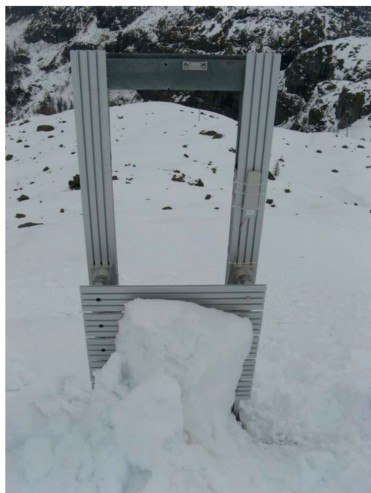

**Figure 9.** Deposition against the obstacle, due to the spontaneous avalanche on 17 April 2013. Photo E. Bovet on April 24.

### 3.1.4. The Avalanche on 20 January 2014

On 20 January 2014, two dry snow avalanches were artificially released; they were mostly dense flows with a little powder component. The avalanches stopped along the slope at about 2375 m asl without reaching the ski run (Figures 2 and 10). As can be inferred by the video analysis, the first avalanche generated two flows: The first one impacted the obstacle with two waves at 2 s interval, while the second stopped before reaching the obstacle. The second avalanche, which was released 14 s after the first one, flowed more on the left of the slope and impacted only marginally the obstacle.

The wedge left by the avalanches in front of the obstacle presented the shape shown in Figure 11 and an average snow density of 220 kg/m$^3$. The total impact forces (calculated considering all the sensors placed on the obstacle as in Figure 1) presented three peaks of 600, 2150, and 1300 N (raw data).

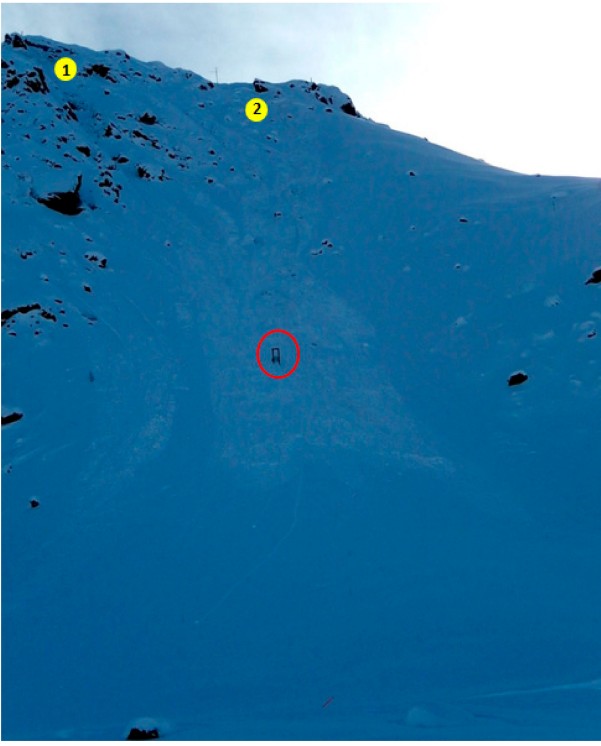

**Figure 10.** Overview of the 20 January 2014 avalanche. The yellow dots indicate the two triggering points and the red circle the obstacle. Photo L. Pitet.

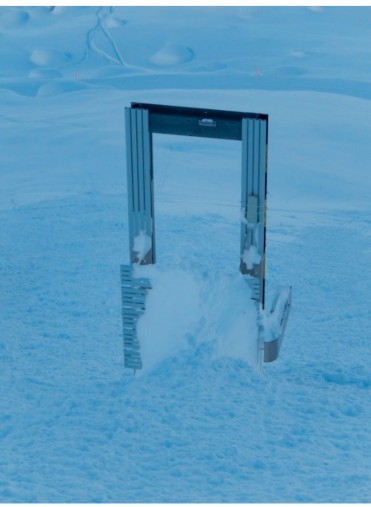

**Figure 11.** Deposition upstream the obstacle left from the avalanche on 20 January 2014. Photo E. Bovet.

### 3.1.5. The Avalanche on 30 April 2014

In the days 27–29 April, the precipitation brought a cumulated new snow sum of 105 cm in three days (AWS Gressoney-L.T.–Gabiet) with a 0 °C level variable between 1900 and 2300 m asl. Therefore, at the test site, the precipitation was a mix of snow and rain, as also observed on the snow surface during the field campaign made on 1 May. A peak in the air temperature of 20 °C on 29 April was recorded at noon at the AWS. On 30 April, the obstacle recorded the impact pressure of an avalanche, which was observed on the next day. It was a wet snow avalanche with an average density in the deposition zone of about 300 kg/m$^3$. The traces of several other avalanches were observed on 1 May. All the events, including the one recorded by the instrumented obstacle, can be classified as loose-snow avalanches. The traces of these events are visible in Figure 12, where the red line refers to the avalanche whose impact forces were measured, according to field observations on the characteristics of the different deposits (see also Figure 2). The total force showed a maximum of 7000 N. The forces varied sharply up to 11,000 N for the two upper plates, while assumed negative values on the lower ones (corresponding to tensile forces upwards the slope). Above the obstacle, with the three lower plates totally submerged in the deep snow cover (HS = 169 cm at the AWS), instead of a proper snow wedge, a snow agglomerate of about 1 m$^3$ with a density of 400 kg/m$^3$ was observed (Figure 13).

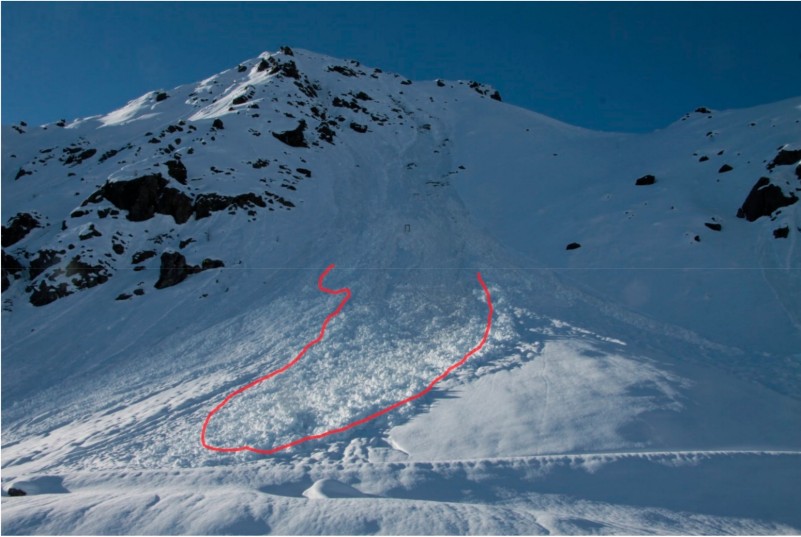

**Figure 12.** Overview of the events observed on 1 May 2014. Among the avalanches released in the period 27–30 April, the red line highlights the deposit of the event on 30 April 2014. Photo M. Maggioni.

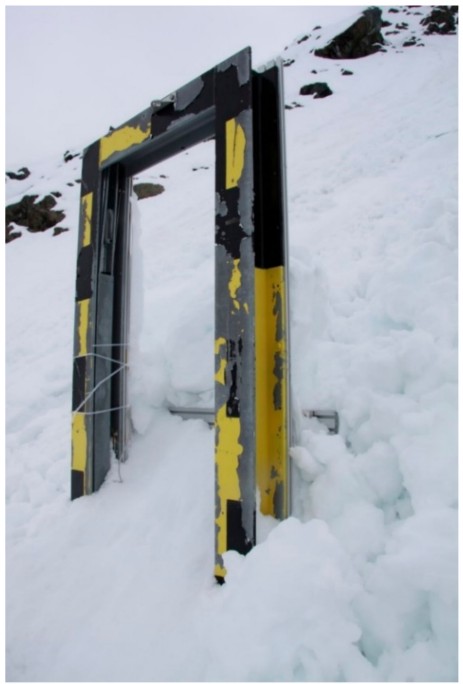

**Figure 13.** View of the obstacle after the event on 30 April 2014 (photo M. Maggioni on May 1). It is clearly visible that the plates are totally submerged in the snow and the large snow agglomerate against the structure.

### 3.2. Impact Pressure Data

For the different selected events, the load transduceres installed on the obstacle, with the different configurations shown in Figure 1, measured the impact forces of the avalanche flows. The data measured by each transducer, as acquired and averaged over 0.2 s time interval, are presented in the Supplementary Materials (Figures S1–S43), while Figure 14 shows the total pressure averaged with moving mean over 0.2 s for the five avalanches.

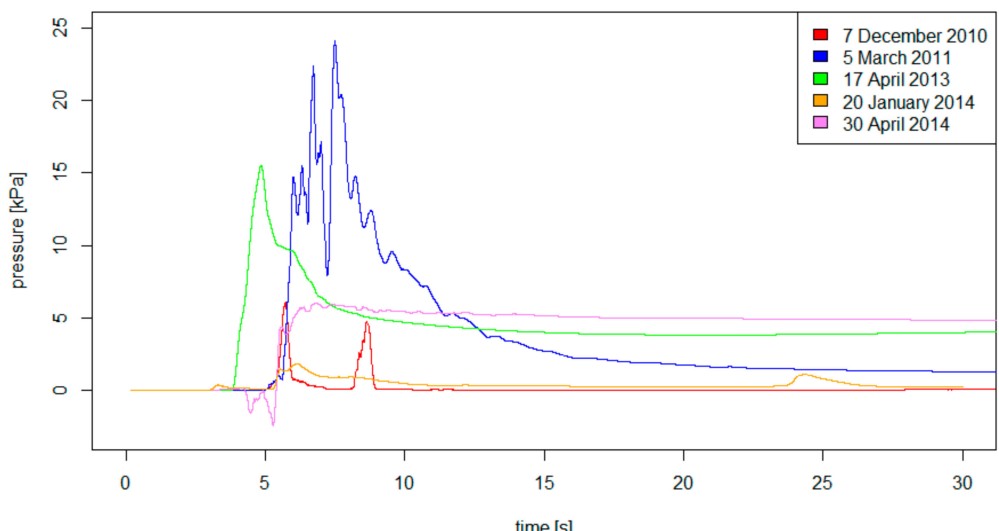

**Figure 14.** Total impact pressures for the five selected avalanches (values averaged over 0.2 s). The beginning of the measurements does not represent the arrival time of the avalanches at the obstacle, as the triggering of the measuring system might have been different for the five events. Figure S44–S48 in the Supplementary Materials report the total force for each avalanche as raw data and averaged over 0.2 s.

The maximum pressure (raw data) measured for the selected events spanned over an order of magnitude, between values of 2 kPa (20 January 2014) and 30 kPa (5 March 2011) (Figures S47 and S45 in the Supplementary Materials, respectively). Those two avalanches were very different in terms of dimensions and snow characteristics (Table 1 and Figure 2). The former stopped along the path just below the obstacle still on a slope angle of about 30° with a snow deposition depth of about 70 cm; the surface snow cover layer had a very low density (60 kg/m$^3$). The latter flowed almost to the ski run with a snow deposition depth up to 1.5 m and developed a powder component, which overrun the ski run; the snow cover presented a surface layer with density of 270 kg/m$^3$. To our knowledge, the event of 20 January 2014 was one of the smallest events measured in a full-scale avalanche test site.

There are clear differences between the time histories of the forces measured for wet avalanches (e.g., 17 April 2013) and the ones obtained for drier events with a powder cloud development (e.g., 7 December 2010 and 5 March 2011). Signals from the load transducers were much more oscillating for powder events than for wet ones (see Figures S1–S43 in the Supplementary Materials). Also, the vertical profiles of the forces were different for powder and wet events.

The vertical profile of the pressures for the wet avalanche on 17 April 2013 was proportional to the depth (Figures 15–17), similar to a hydrostatic contribution, while for the other events, the pattern is more complex, indicating that both flow density and velocity play a role in determining the pressures. The small dry avalanche on 20 January 2014 presented pressures that are vertically distributed in a variable manner, tending to be concentrated on the upper plates (Figures 15–17). This might be an indication of an avalanche flow in a cold dense flow regime, which generated pressure driven more by velocity than by density. Though, as we did not measure such variables, we are aware that other facts might explain this pressure distribution. As at the test site we did not measure flow depth, we cannot know the level of the sliding surface. On 20 January 2014, the snow depth measured at the near AWS of Lake Gabiet was 129 cm (Table 1). Thus, the highest pressures on the upper plates might be also related to the higher position of the sliding surface. From field work, however, we found traces of the passage of the avalanche also in correspondence of the lower plates. Thus, we think that our indication of a cold dense flow regime is correct.

The powder avalanche on 7 December 2010 presented a hydrostatic profile for the first impact while for the second impact, the profile presented a maximum on plate II (Figure 16). For this event, the lowest measuring plate was 110 cm above the ground, therefore it is possible that the densest core flowed below the plates and only marginally interacted with the lowest one, probably during the first impact. Instead, during the second impact, velocity was more important in determining the impact pressures. As we did not measure density and velocity of the flow, we are aware that these are only suppositions that cannot be proved, however they give insights on how avalanches are complex evolving phenomena whose characteristics can change dramatically during the flow.

A good example of the presence of different regimes within an avalanche flow is the event on 5 March 2011. The evolution of the pressures for such event suggests that an intermittency frontal zone was present in the front of the avalanche before the dense flow component impacted the obstacle. In fact, the recorded forces at the beginning of the impact showed large variations (as typical of the intermittency zone—see [18]): In particular, the upper transducers (numbers 7, 8, and 9) measured values oscillating from 0 to 4000 N during the first 2 s of flow (see Figures S11–S13 in the Supplementary Materials). Even for this event, as in 2010, the lowest measuring plate (II) was placed at 110 cm above ground, therefore we are not able to describe the vertical profile of the pressures in a complete way. However, from the available measurements, it seems that the centroid of the forces was most of the time concentrated on plates III and IV.

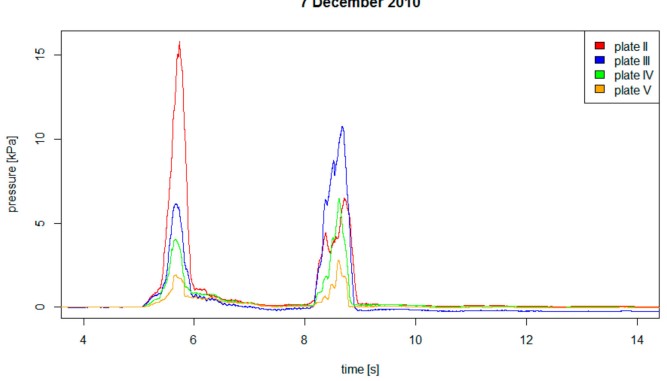

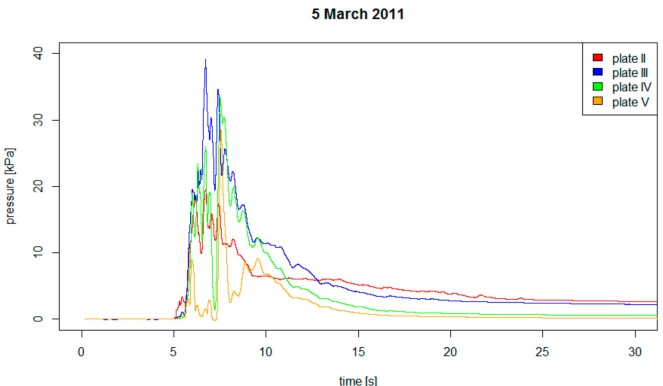

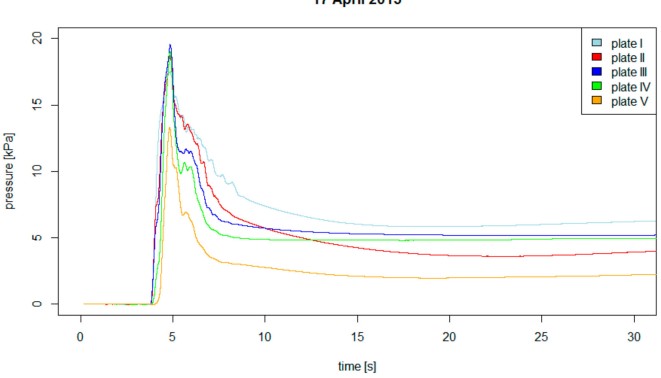

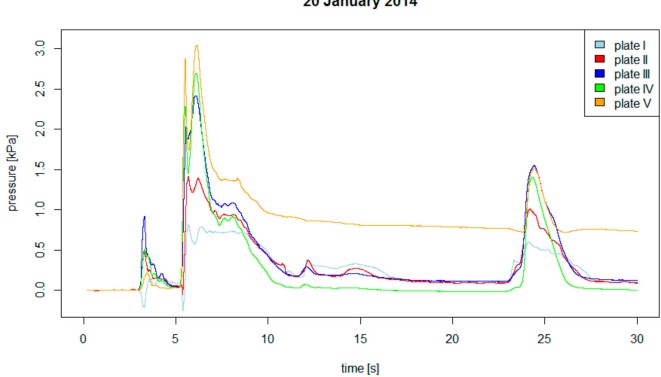

**Figure 15.** *Cont.*

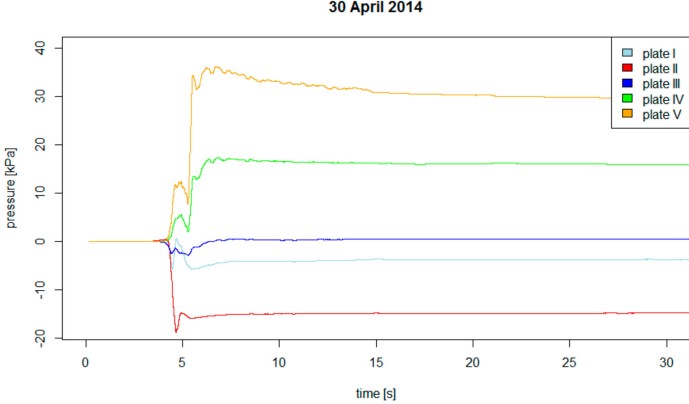

**Figure 15.** Time evolution of the pressure on the different plates for the five selected avalanches (data averaged over 0.2 s).

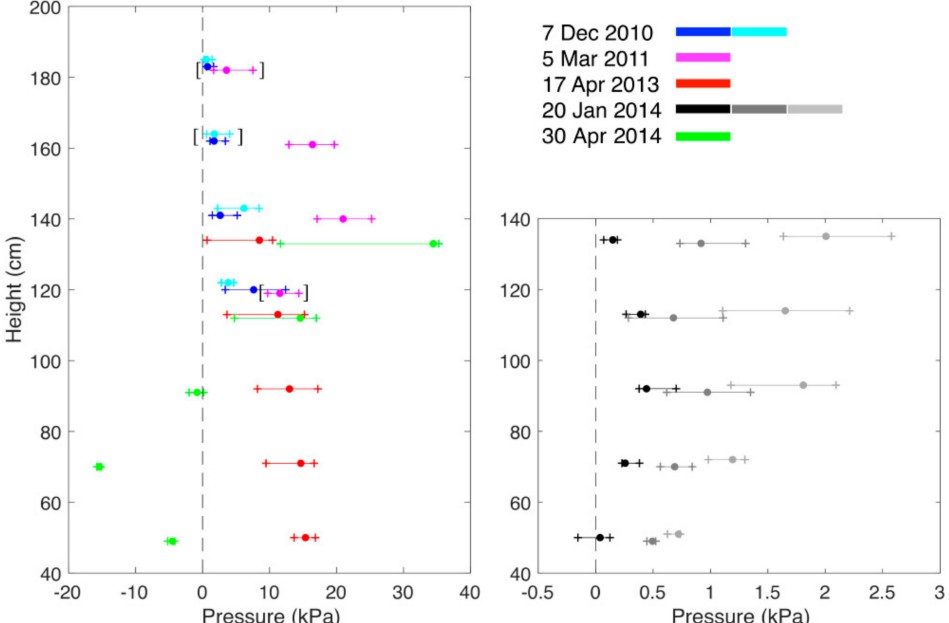

**Figure 16.** Vertical profiles of the pressures during the time intervals when the impacts occurred, for the five selected avalanches (data averaged over 0.2 s). The considered time intervals shown with colors are: 5.40–6.00 s (blue) and 8.10–8.90 s (light blue) for 7 December 2010; 5.65–9.20 s (pink) for 5 March 2011; 3.90–5.40 s (red) for 17 April 2013; 3.15–3.76 s (black), 5.35–7.30 s (grey), and 23.75–26.36 s (light grey) for 20 January 2014; 4.25–7.70 s (green) for 30 April 2014. The symbols show the median values within the considered time intervals and the bars represent the 25th and 75th percentiles. In squared brackets are the values related to inoperative sensors (see text for details).

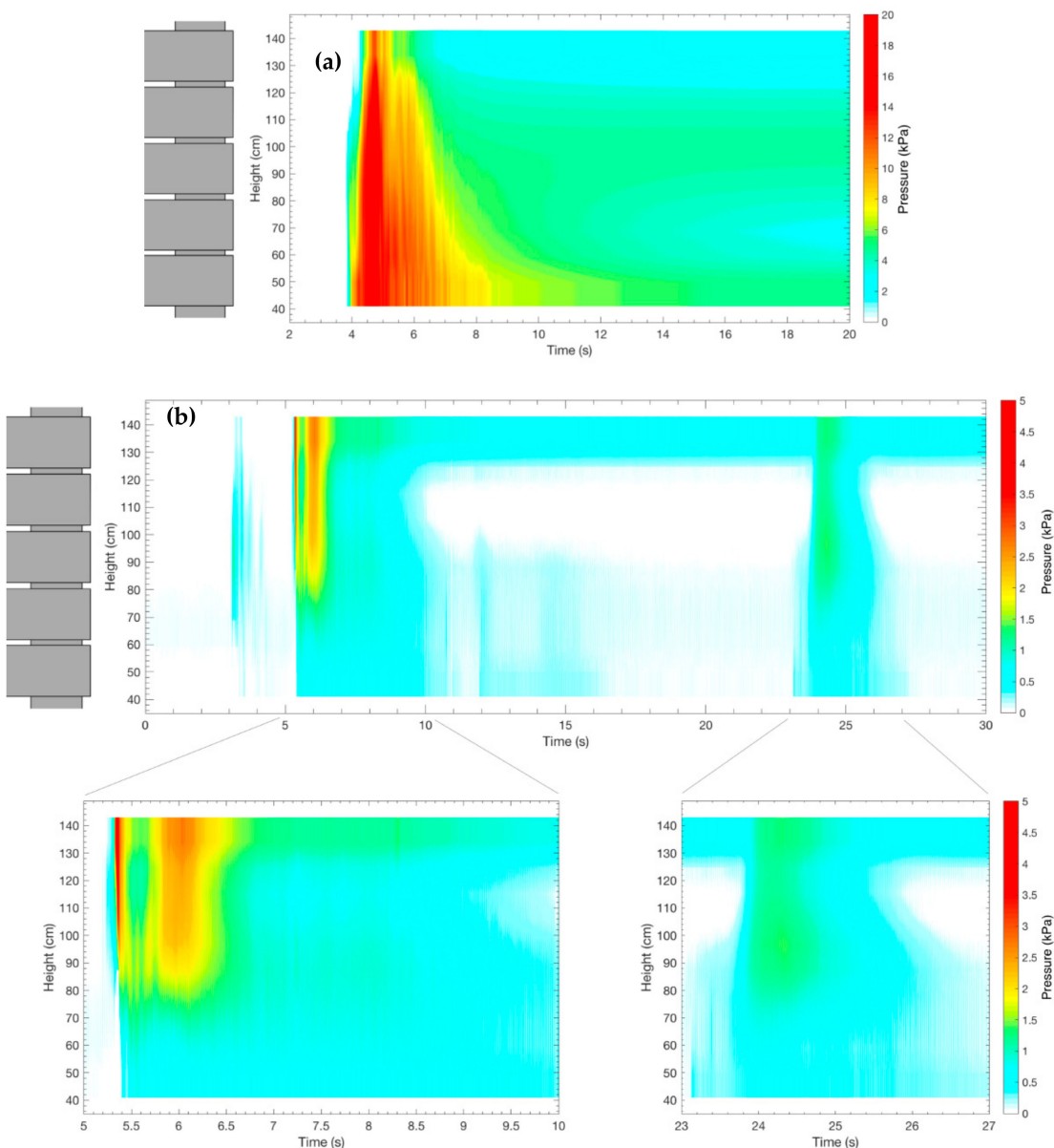

**Figure 17.** Pressure distribution for the (**a**) 17 April 2013 and (**b**) 20 January 2014 avalanches (data averaged over 0.2 s).

The analyses of the position of the centroid of forces give information on where the resultant force was concentrated over the impact area. The time history of the position of this point describes an interesting aspect of the evolution of the interaction between the flowing snow and the obstacle, highlighting different phases of the impact, each characterized by a prevalent loading mode.

For the events on 20 January 2014 and 17 April 2013, the impact area was about 1.13 m² as all the cells were working (Figure 1). For the wet snow avalanche on 17 April 2013, the center of forces moved within a small area of 90 mm x 150 mm on the lower part of plate III (Figure 18a). After the passage of the avalanche, the center of the forces moved horizontally from the right-hand side almost to the center. Those forces, measured at the end of the event, were not related to the avalanche flow anymore, but to the presence of the snow wedge, which presented a symmetric shape located a little to the right-hand side of the obstacle (see Figure 9).

For the small dry avalanche on 20 January 2014, the center moved within a wider area of 250 mm × 400 mm. When the first wave of the first avalanche impacted the obstacle, the force was concentrated on plate III (single dark dot (0,100) in Figure 18b), then it remained on plate II between the two waves

and then moved up on plate III during the impact of the second wave of the first avalanche; later, it was again on plate II during the impact of the second avalanche. The position of the center high up on plate IV on the right-hand side of the impacted area is related to the values of about 150 kN registered by plate V in between the two avalanches (see also Figure 15 and Figures S32 and S33 in the Supplementary Materials).

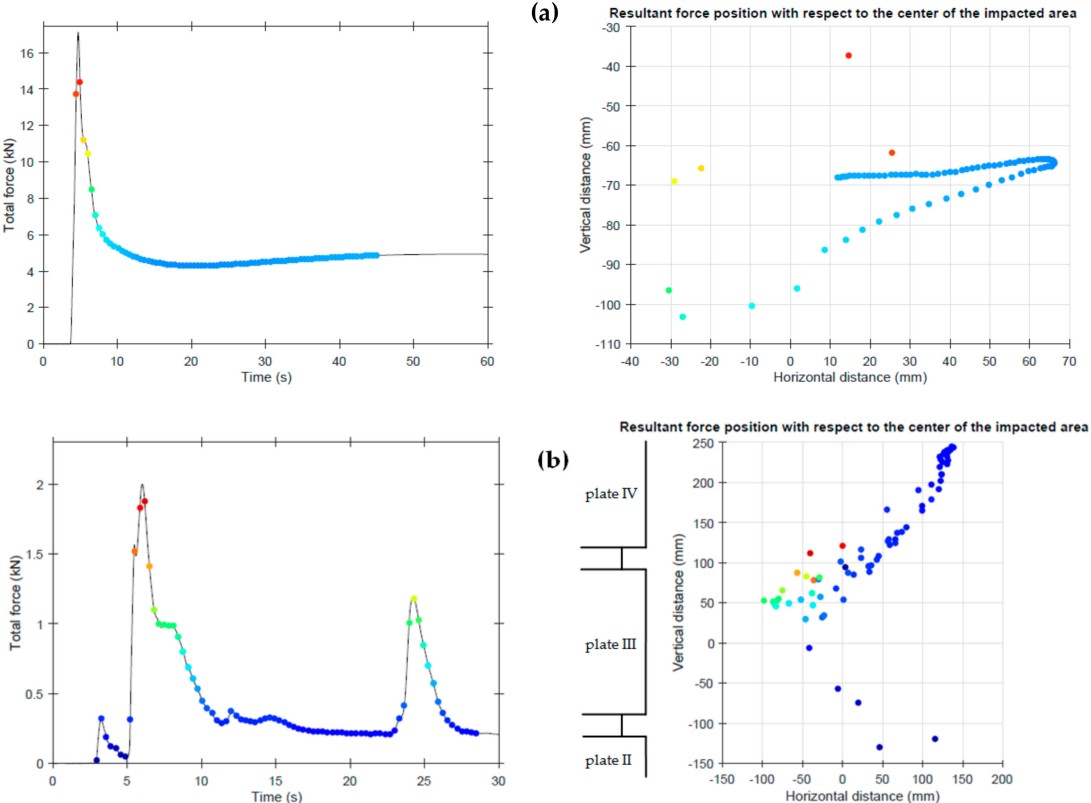

**Figure 18.** Center of forces for the (**a**) 30 April 2013 and (**b**) 20 January 2014 avalanches (data averaged over 0.2 s). The colored dots help the reader to follow the time evolution from plot (**a**), where the time is on the abscissa, to plot (**b**), where the time is not represented.

After the avalanches stopped, the snow wedge generated a static load on the obstacle, which is well visible in the force measurements. On 5 March 2011, a load of about 1500 N was recorded on the two lower plates after the avalanche. On 20 January 2014, the static load was much lower (150 N). The density of the snow wedge was 320–360 kg/m$^3$ in the first case and only 220 kg/m$^3$ in the second one.

As we noticed an asymmetric shape in the snow wedge (Figure 6, Figure 9, and Figure 11), we thought that this asymmetry might be found in the measured forces, with lower impact forces on the plates covered by the snow wedge. In actuality, no large differences were found between forces measured on the right-hand and left-hand side of the obstacle (see Figures S8–S43 in the Supplementary Materials).

During the event registered on 30 April 2014, when the obstacle was almost completely submerged into the thick snow cover, the two upper plates measured a sharp increase in the forces, which remained almost constant (3500 and 7000 N for plate IV and V, respectively) until the end of the recording (Figure 15 shows the corresponding pressures). During the post-event field work, we found a snow agglomerate of 1 m$^3$ with density of 400 kg/m$^3$ against the obstacle (Figure 13). We think that this is the reason that explains the measurement of forces that do not decrease to zero after the event. However, more intriguing is the data measured from the lower two plates, which show negative values.

Negative data in the forces, which were sometimes measured especially at the beginning of the avalanche impact, might be due to the assumption that no force was applied when measurements started, resetting the measured values from each transducer to zero before the event. In reality, if some snow was present against the plates before the avalanche impact, this would result in negative values related to the unloading caused by erosion of the pre-existent snow from the frontal part of the avalanche. This occurred, for example, on 20 January 2014, when some snow deposited upstream from the obstacle was observed before the artificial triggering: The forces on plate I presented, in fact, negative values at the very beginning of the avalanche flow (Figures S24 and S25 in the Supplementary Materials). However, this is not the case of the avalanche on 30 April 2014, when the lower plates were totally submerged in the thick, dense, and wet snow cover.

## 4. Discussion

The data recorded at the Seehore experimental test site, which we presented in this paper, describe avalanches of small to medium size with a wide range of snow characteristics (from very cold and dry to wet snow conditions). The peaks in the impact pressures were not simply related to the avalanche dimension. For example, the largest powder snow avalanche on 7 December 2010 generated lower impact forces than the one occurred on 5 March 2011 (same configuration for the load transducers). In fact, though initially the avalanche potential damages were classified according to the avalanche dimensions [19], it is now well known that impact pressures are not only related to avalanche dimension, but also to snow characteristics and flow regimes (e.g., [20,21]). The event on 20 January 2014 was a combination of characteristics, which generated very low impact pressures at the obstacle: Small avalanche, very low snow density, and cold temperatures.

In the context of the existing experimental test sites, avalanches at the Seehore test site generated maximum impact pressures ranging from 2 to 30 kPa. The Col du Lautaret test site, which is very similar to the morphological characteristics to our test site (though with longer runout distance), is equipped with strain gauges, which allowed pressure measurements up to 5260 kPa. However, avalanches were generally of small to medium dimension with typical pressures of a few hundred kPa [5]. In larger test sites, such as Vallée de La Sionne and Ryngfonn, the impact pressures can reach much higher values. In Vallée de La Sionne, measured pressures went up to more than 1000 kPa [21]. Reference [3] reports average values of hundreds of kPa at the beginning of the runout zone at the Ryggfonn test site, but with maximum values up to 720 kPa.

From the analyses of the pressure distribution and, when available, of other dynamical variables, some information on the avalanche flow regimes might be derived. Köhler et al. [7] defined seven different flow regimes revealing that small- to medium-sized avalanches tend to be simple with only one flow regime, while larger avalanches are complex and exhibit multiple flow regimes. Our data recorded for the avalanche on 5 March 2011 seem to also show that small avalanches can present different flow regimes. It seems that on 5 March 2011, a frontal intermittency region in the first 2 s of the impact was present, followed by a dense flowing part.

For some events, a snow wedge was formed against the obstacle. The snow wedge surely influenced the registered impact forces and the related pressure due to the different shape in front of the cells [21,22]. Bovet et al. [23] analyzed the interaction between the avalanche flow of 5 March 2011 and the obstacle with FEM models and found a ratio of 1.37 between the impact pressures on a snow-free surface (flat area) and the ones on the snow wedge (dihedral shape). Thibert et al. [22] found a relationship between the pressure measured on a plate-like obstacle and on cantilever sensors placed at the Col du Lautaret test site: Generally, the cantilever signals overpassed the plate pressure signal by a ratio within a range of 1.6 to 2.9. Similar findings were also found by [24] for larger events in Vallée de La Sionne.

The above-mentioned fact suggest that recorded pressures by plate-like obstacles where a snow wedge can easily form should be carefully used in the calibration of dynamical models that do not include the interaction of the avalanche flow with obstacles. It is not the aim of this paper to discuss

how to calculate the impact pressure from the dynamical variables of an avalanche flow, but we are aware of the importance of a complete set of measurements to try to give an answer to this issue. In fact, the large difference between Seehore test site and Norwegian, French, and Swiss ones is that we do not measure flow depth, velocity, and density, therefore we cannot truly discuss our pressure measurements with respect to the common way of calculating pressures in case of the different flow regimes. Also, the position of the center of the forces and the peak pressure on the impacted area should be analyzed with respect to the flow depth and the position of the sliding surface. At Ryngfonn, for example, Norem et al. [25] found a vertical distribution in the pressure with peak values on the upper load cells, which were clearly related to the observed snow cover conditions.

The pre-existing snow cover conditions should also be taken into consideration in the pre-processing of the pressure data. From our experience, the resetting of the data to zero before the event is correct in the case of no snow accumulation against the obstacle before the avalanches. Instead, when some load transducers are within the natural snowpack or are covered by a snow wedge formed by previous avalanches, this resetting might be not correct and may cause loss of information. The event of 30 April 2014 clearly shows how this resetting produced negative forces on the lower cells, which are not physically explainable and highlights the importance of considering the pre-existing snow cover condition in the analyses of the impact pressure. This is particularly true for small avalanches.

## 5. Conclusions

In this work, we analyzed five selected avalanches, which occurred at the Seehore test site. The aim was to present the impact pressures recorded on the instrumented obstacle and discuss those data with respect to the snow characteristics. To summarize, we found that small- to medium-sized avalanches generated impact pressures in the range of 2 to 30 kPa. The vertical distribution of the impact pressures suggests that different flow regimes occurred in the different avalanches. Such differentiation was also observed within each single event. In the analyses of the data we strongly realized that the pre-existent snow cover conditions should be considered in the choice of the pre-processing of the data, in order, for example, to correctly consider erosion processes in front of the obstacle at the beginning of the impact.

We are aware that only with a complete set of measurements and with more sophisticated techniques, a deep insight into the avalanche dynamic, especially concerning the interaction with an obstacle, is possible. We believe anyway with this work to add some useful data to the ongoing discussion on snow avalanche impact on obstacles.

**Supplementary Materials:** The following are available online at http://www.mdpi.com/2076-3263/9/11/471/s1, Figure S1. 7 December 2010: impact forces recorded by load transducer n. 03; Figure S2. 7 December 2010: impact forces recorded by load transducer n. 04; Figure S3. 7 December 2010: impact forces recorded by load transducer n. 05; Figure S4. 7 December 2010: impact forces recorded by load transducer n. 07; Figure S5. 7 December 2010: impact forces recorded by load transducer n. 08; Figure S6. 7 December 2010: impact forces recorded by load transducer n. 09; Figure S7. 7 December 2010: impact forces recorded by load transducer n. 10; Figure S8. 5 March 2011: impact forces recorded by load transducer n. 03; Figure S9. 5 March 2011: impact forces recorded by load transducer n. 05; Figure S10. 5 March 2011: impact forces recorded by load transducer n. 06; Figure S11. 5 March 2011: impact forces recorded by load transducer n. 07; Figure S12. 5 March 2011: impact forces recorded by load transducer n. 08; Figure S13. 5 March 2011: impact forces recorded by load transducer n. 09; Figure S14. 17 April 2013: impact forces recorded by load transducer n. 01; Figure S15. 17 April 2013: impact forces recorded by load transducer n. 02; Figure S16. 17 April 2013: impact forces recorded by load transducer n. 03; Figure S17. 17 April 2013: impact forces recorded by load transducer n. 04; Figure S18. 17 April 2013: impact forces recorded by load transducer n. 05; Figure S19. 17 April 2013: impact forces recorded by load transducer n. 06; Figure S20. 17 April 2013: impact forces recorded by load transducer n. 07; Figure S21. 17 April 2013: impact forces recorded by load transducer n. 08; Figure S22. 17 April 2013: impact forces recorded by load transducer n. 09; Figure S23. 17 April 2013: impact forces recorded by load transducer n. 10; Figure S24. 20 January 2014: impact forces recorded by load transducer n. 01; Figure S25. 20 January 2014: impact forces recorded by load transducer n. 02; Figure S26. 20 January 2014: impact forces recorded by load transducer n. 03; Figure S27. 20 January 2014: impact forces recorded by load transducer n. 04; Figure S28. 20 January 2014: impact forces recorded by load transducer n. 05; Figure S29. 20 January 2014: impact forces recorded by load transducer n. 06; Figure S30. 20 January 2014: impact forces recorded by load transducer n. 07; Figure S31. 20 January 2014: impact forces recorded by load transducer n. 08; Figure S32. 20 January 2014: impact forces recorded by load transducer n. 09;

Figure S33. 20 January 2014: impact forces recorded by load transducer n. 10; Figure S34. 30 April 2014: impact forces recorded by load transducer n. 01; Figure S35. 30 April 2014: impact forces recorded by load transducer n. 02; Figure S36. 30 April 2014: impact forces recorded by load transducer n. 03; Figure S37. 30 April 2014: impact forces recorded by load transducer n. 04; Figure S38. 30 April 2014: impact forces recorded by load transducer n. 05; Figure S39. 30 April 2014: impact forces recorded by load transducer n. 06; Figure S40. 30 April 2014: impact forces recorded by load transducer n. 07; Figure S41. 30 April 2014: impact forces recorded by load transducer n. 08; Figure S42. 30 April 2014: impact forces recorded by load transducer n. 09; Figure S43. 30 April 2014: impact forces recorded by load transducer n. 10; Figure S44. 7 December 2010: total force on the impacted area; Figure S45. 5 March 2011: total force on the impacted area; Figure S46. 17 April 2013: total force on the impacted area; Figure S47. 20 January 2014: total force on the impacted area; Figure S48. 30 April 2014: total force on the impacted area.

**Author Contributions:** Conceptualization, M.M., M.B., F.B., B.F., M.F., B.C., V.D.B. and O.P.; validation, O.P., M.M., M.B.-B.., V.D.B., and F.B.; formal analysis, M.M. and V.D.B.; investigation, all authors and E.B., L.P., D.V., and A.W.; data curation, O.P. and M.M.; writing—original draft preparation, M.M. and M.B.-B.; writing—review and editing, all authors; visualization, M.M. and V.D.B.; supervision, B.C., M.F., and V.S.; project administration, B.C., B.F., M.F., L.P., and V.S.; funding acquisition, B.C., B.F., M.F., and V.S.

**Funding:** This research was funded by Regione Autonoma Valle d'Aosta in the framework of DYNAVAL and MAP3 projects—Operational programme "Italy–France (Alps—ALCOTRA) 2007–2013".

**Acknowledgments:** We thank the Regione Autonoma Valle d'Aosta, owner of the Seehore test site, in particular Valerio Segor (V.S.); Eloise Bovet (E.B.), Luca Pitet (L.P.), Davide Viglietti (D.V.) and Danilo Godone for support in field works and for interesting discussions; Monterosa S.p.a. for the logistic support, in particular Arnoldo Welf (A.W.) and Henry Grossjacques.

**Conflicts of Interest:** The authors declare no conflict of interest. The funders had no role in the design of the study; in the collection, analyses, or interpretation of data; in the writing of the manuscript, or in the decision to publish the results.

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
