# Peer review of "Snow Avalanche Impact Measurements at the Seehore Test Site in Aosta Valley (NW Italian Alps)"

_geosciences, doi:10.3390/geosciences9110471_

Round 1

Reviewer 1 Report

This is a useful contribution, with very good documentation of the impact from 5 small avalanches. However, it would be more useful if the discussion - if possible - compared the findings to published characteristics of avalanches in motion (e.g. The design of avalanche protection dams). For example:

Is the flow height of the dense core comparable to other sources, e.g. 1-2 m in unconfined terrain? is there evidence of a saltation (fluidized) layer ahead of, or above, the dense core? Did the wet avalanche move as a plug?

Minor specific comments:

The captions to Figs. 14 to 17 should provide some explanation or interpretation to give the figures stand-alone value.

Line 131. Suggest "each about 1 s long and 2 s apart"

Line 319 Explain intermittency. I think it means large variations in impact pressure.

Line 356. "Wedge" is an excellent descriptive word. "Dihedral" does not add to the description. Very minor point.

Line 370, Caption to Fig. 17. What do the colors mean? Same as Fig. 16?

Line 450 Replace "loose" with "lose"

Reviewer 2 Report

I think the general idea of the paper is sound, but it lacks rigor. A few examples are simple, which is where I will start.

16,17: The avalanches are said to be "small to medium-sized", this statement is vague. If possible use some quantifiable values such as the European classification system. If this is what you did then indicate as such with a reference.

31: It is stated that "additional literature exists", it should be listed.

61: The "Materials and Methods" should not include any discussion of results, it should only covered how and what was measured and what calculations were performed on the recorded data. Everything from Sec. 2.3 onward should be in Sec. 3.

78: There should be a brief discussion regarding when the plates were moved. This might be in the references listed on 76, but it is important enough to include again.

249: It is stated that "some transducers were sometimes not operative." How was this determined? Is it possible for the sensor to become buried and no longer register data? Did the pressure exceed the limits of the sensor? Was the sensor above the flow?

250: How was it determined to be reasonable to estimate "not operative" values as the average of the other values? I fundamentally disagree with this approach, especially since you report vertical pressure profile data. At a minimum the points should be interpolated with some sort of polynomial
function that accounts for depth. Consideration should also be given, as mentioned above, to plates that may of been buried or above the flow. Also, all results showing estimated data should indicated
as such. For example, Fig. 14 should mark the line segments that were estimated in some fashion so that the
results are unambiguous.

262: Analysis was performed for "some significant instants", which is totally meaningless. How is the data significant? How was it determined to be significant? Why was that measure of significance used? With respect to the results presented in Fig. 15 there is no need to only select certain times. Color map plots such as in Fig. 16 could be used thus eliminating the need to capture snapshots.

As a reviewer, the above simple examples demonstrate that more rigor is needed to be certain the data and conclusions presented are accurate and useful to the community. As such, I will not spend time assessing results when I am uncomfortable with the methods.

On a larger and more general level I feel the research presented is contrary to the stated purpose in the paper: "that the data...might add some useful information to the ongoing scientific discussion..."

On line 100 it is stated that 63 avalanches were recorded, which is wonderful, but only the results of a hand-picked set of 5 is presented. The reason for the selection of the 5 is given on line 104 as to cover the "different ranges of force measurements and avalanche types". Such a selection lacks scientific merit and intentionally skews the obtained results to an arbitrarily selected subset of data.

As such, as a reviewer I feel that all 63 should be presented in this paper and that raw data be made available for interested researchers. Of course it is not possible to provide detailed descriptions of all 63 events, but it is certainly possible to present aggregate findings such as done in Table 1.
I also feel that it is reasonable to present pressure plots, such as in Fig. 14, for each as well. This general overview of all events would be the "results" section and should be the bulk of the paper if the stated purpose is in fact to provide "useful information to the ongoing scientific discussion..."

Then in the discussion (Sec. 4) your contributions to the aforementioned "scientific discussion" can be presented. The 5 specific events could be disused including evidence that these events do cover the range of
data recorded. Additionally, the discussion could try to summarize the average pressure, max pressure, duration, etc. based on any number of factors presented in Table 1. For example, aggregate the pressure data for all
dry slabs and present the expected range of pressures based on statistical methods.

As mentioned, I feel that the work presented lacks the rigor necessary for publication. In particular, I find that hand-picked results to be most troubling. That being said, I hope that my comments encourage the authors
to continue their hard work and compile a more complete picture of the rresearch completed. If this is done then I feel strongly that the research presented will be of great benefit to the scientific community.

Round 2

Reviewer 2 Report

My comments have been addressed, I support publication. Thank you for your patience with all my silly technology problems.